**Perspective**

# Applying a causal biopsychosocial model to guide medicine and psychiatry fifty years after Engel

Roland von Känel ✉

Fifty years after George L. Engel proposed the biopsychosocial model to replace biomedical reductionism, its influence across medical disciplines, including psychiatry, endures but its scientific potential remains unrealized. Conceived as a foundation for research, education and clinical practice, Engel's model anticipated the current search for integrative explanations linking biological, psychological and social processes. Yet in practice, it has often become a checklist rather than a causal framework. Contemporary tools, including computational modeling, network analysis, digital phenotyping and causal inference methods, enable increasingly sophisticated modeling and partial testing of these interactions across different levels of organization. This Perspective calls for a causal biopsychosocial framework to guide the next fifty years of research, teaching, and patient care that integrates measurable and predictive sciences of interacting systems to unite medicine and psychiatry.

## From blueprint to rhetoric

In 1977, George L. Engel published his landmark article in *Science* calling for a biopsychosocial (BPS) model to replace narrow biomedical reductionism[1]. He argued that disease arises from dynamic interactions among biological, psychological, and social systems, using general systems theory to describe a causal hierarchy linking molecules to societies. Illness, in this view, reflects perturbations that propagate through these strata and bring molecular, mental, and social processes into interaction. Today, such multilevel interactions are increasingly open to mechanistic and empirical investigation. Yet, the BPS model has often been reduced to a checklist exercise, with clinicians dutifully noting "biological," "psychological," and "social" factors without articulating how they interact[2]. One reason this occurs may be methodological. Biological mechanisms became experimentally tractable earlier than psychosocial interactions, which encouraged reductionist default models. In the absence of formal tools to model cross-level causation, clinicians often documented domains without specifying how these interacted. Critics have therefore dismissed the BPS model as vague or eclectic[3], while others think it only has relevance to psychiatry[4], despite Engel's explicit aim to transform medicine as a whole[5]. These misinterpretations obscure Engel's original intent, which was to advance a rigorous, causally integrated science of health and disease. As Freud once anticipated that biology would eventually reshape psychological theory[6], Engel foresaw medicine causally linking rather than segregating biological, psychological, and social explanations.

The enduring strength of the BPS model lies precisely in this ongoing effort to integrate all explanatory levels through regulatory causation across biological, psychological, and social domains[7]. However, to fully realize the model's scientific potential, there is a critical need for mechanistic studies that explore how these causal relationships unfold in real-world contexts, mapping the specific pathways through which biological, psychological, and social factors interact and regulate one another. In this Perspective we argue that Engel's model should now be operationalized as a causal biopsychosocial framework for specifying, measuring, and testing how biological, psychological, and social processes interact across levels of organization to shape disease, illness, and care.

## Achievements of the model

Where Engel's vision has been applied with empirical rigor, the BPS model has proven remarkably productive. Across medicine and psychiatry, integrative research has revealed links between psychological and social processes and biological mechanisms, from inflammation and autonomic regulation to neural plasticity and treatment response. In chronic inflammatory disease, the BPS framework clarifies causal sequencing. As an example, patients with inflammatory bowel disease show a 1.5-fold increase in the prevalence of depressive symptoms during flare-ups[8], mediated by

Department of Consultation-Liaison Psychiatry and Psychosomatic Medicine, University Hospital Zurich, University of Zurich, Zurich, Switzerland.
✉e-mail: roland.vonkaenel@usz.ch

proinflammatory cytokines that disrupt mood and sleep[9]. Likewise, up to 40% of cancer patients treated with interferon-α develop major depression[10]. Here, biology initiates the cascade, where immune activation induces psychological change, which then feeds back into the disease course[11]. Conversely, psychosocial interventions can enhance immune function and dampen inflammation, underscoring the bidirectional nature of these effects[12]. Moreover, social stressors and depressive states themselves can trigger inflammatory responses, thus inflammation is not always causal but should be viewed as a key mediator linking social context, psychological processes, and disease[13–15]. This bidirectional causality, whether inflammation drives depression, depression drives inflammation, or both simultaneously, has been central to recent debates in psychoneuroimmunology, including whether inflammation markers in depressed patients reflect a casual pathway or are epiphenomena of comorbid medical conditions and unhealthy lifestyle[13,16].

Psychosocial factors can also act first. Anger or grief may trigger acute myocardial infarction[17] or Takotsubo cardiomyopathy[18] within hours. In Takotsubo syndrome, acute emotional stress drives sympathetic surges, marked by norepinephrine release, that transiently stun the myocardium, predominantly in postmenopausal women[19]. A longitudinal study showed that perceived stress predicted amygdalar activation, bone marrow stimulation, arterial and systemic inflammation, and subsequent cardiovascular events[20], illustrating how subjective experience can prospectively predict and be associated with measurable biological risk.

The social context itself can be causal. Placebo and nocebo research shows that expectations, trust, and the clinician's words shape pain perception through defined neurobiological pathways. Functional magnetic resonance imaging studies show prefrontal modulation of brainstem pain circuits[21]. Placebo analgesia recruits endogenous opioids and dopamine[22], whereas nocebo hyperalgesia is mediated by cholecystokinin[23]. Social influences also operate at a structural level and can shape biological outcomes directly, independent of individual psychological mediation. In a large population-based cohort, low social integration and limited financial support predicted incident cardiovascular events and mortality even after adjustment for socioeconomic indicators, health behaviors, biological risk factors, and depression[24]. In a population-based diabetes cohort, neighborhood deprivation predicted cardiometabolic risk, including higher body mass index and poorer glycemic control, independent of individual-level income and education[25].

Together, these findings exemplify Engel's core insight that biology, psychology, and social forces are inseparable, each capable of initiating and sustaining feedback loops that shape health and disease. Importantly, in some of these domains the BPS model explicitly motivated integrative hypotheses, whereas in others it has served as a retrospective framework to organize converging findings. Its heuristic value lies not in claiming priority for any level, but in encouraging researchers to formulate multilevel hypotheses a priori, for example by specifying how stress exposure, inflammatory processes, and relational context jointly predict disease course, and by testing such models longitudinally. Used in this way, the BPS model becomes a research strategy rather than a post hoc interpretive lens.

## Criticisms of the biopsychosocial model

Despite its influence, the BPS model has been criticized more than celebrated, especially in psychiatry[26]. Some call it unstructured eclecticism that explains everything and thus nothing[3]. Others argue it overlooks broader social forces such as poverty, social inequality, and political context, underestimating their impact on health[27]. Some biological integrationists emphasize that psychosocial interventions exert effects through measurable neural plasticity, highlighting biological embedding rather than denying psychosocial causation[28]. Phenomenological and enactivist scholars argue that lived experience and relational context are constitutive of biological processes rather than secondary to them[29]. Together these views cast Engel's model as too vague, inclusive, or mechanistically thin.

In medicine, critique is rarer but similar. The model is seldom operationalized in rehabilitation or chronic care, allowing systems to revert to biomedical defaults[30]. It is selectively applied, embraced for disorders with biomedically "unexplained" symptoms, such as non-epileptic seizures, fibromyalgia, or chronic fatigue, but largely ignored in general medicine and surgery[31]. Conceptual analyses highlight confusion between disease (biological dysfunction), illness (subjective experience), and sickness (social role), warning the model can blur distinctions essential to clinical reasoning[32].

These critiques partly hold but misread Engel's aim: integration, not relativism. Frustration lies less with the concept than with its lack of precision. Yet attempts to tighten it often reintroduce reductionism. Those elevating relationships and context merely invert biology's primacy; purists who minimize psychosocial factors ignore evidence these can be proximal disease causes. The issue lies not with Engel's model, but with the insufficient effort across disciplines to rigorously define and specify causal pathways, particularly when complex biological processes are overlooked in favor of philosophical abstractions. In practice, particularly for disorders such as depression, "rigorously defining" causal pathways does not mean isolating a single decisive mechanism, but formulating provisional, multilevel causal schemas that can be empirically tested, revised, and constrained over time. In psychiatry, this gap widened after the unrealized "Decade of the Brain"[33] and subsequent philosophical turns[34], which, while insightful, sometimes deflected attention from measurable mechanisms. The "Decade of the Brain" raised expectations that mental disorders would soon be explained primarily through neurobiology, but its limited clinical translation fostered disappointment with reductionist accounts. Subsequent phenomenological and enactive approaches re-emphasized embodiment, lived experience, and relational context, yet did not always clearly specify which mechanisms should be measured or how cross-level interactions can be empirically tested.

Such a retreat from mechanism risks throwing out the "biopsychosocial baby" with the reductionist bathwater just as the framework matures. Advances in psychoneuroimmunology, social neuroscience, and computational psychiatry now begin to articulate structured, testable models of multilevel interactions. Predictive coding explains functional seizures as maladaptive priors about bodily states[35]; central sensitization accounts for amplified pain in fibromyalgia[36]; and interoceptive inference clarifies why fatigue feels overwhelming without organ pathology[37]. Far from exposing the BPS model's limits, such discoveries show its refinement: functional and mental disorders are not failures of the BPS framework but highlight precisely the kinds of multilevel interactions it was designed to address. In the same spirit, recent work within active inference[38], enactivism[39], and philosophical psychiatry[40] has called for a more causally integrated understanding of the BPS model. Approaches such as symptom inference models, externalist psychiatry, and the cultural-ecosocial systems framework foreground subjectivity, narrativity, and cultural processes as constitutive of mechanisms rather than contextual add-ons[41]. These efforts are complementary to the present proposal, whose goal is not to replace such frameworks but to emphasize shared commitments: explicit modeling of cross-level causation and empirical tractability.

## Learning from misapplications of the model

Recurring misapplications and misconceptions illustrate how the BPS model is misunderstood. The "three boxes" trap treats biology, psychology, and society as parallel domains rather than interacting systems. An oncology patient may have tumor markers (biological), distress scores (psychological), and marital status (social) recorded, yet the links among them remain unspecified. Without causal modeling, the BPS approach devolves into checklists and loses explanatory power.

This emphasis on causal modeling serves all medicine (not just psychiatry), specifying how social stressors predict surgical infections or distress accelerates cancer, without biological reductionism. In fact, some enactivist and relational approaches emphasize that cognition and experience emerge through dynamic interactions between brain, body, and environment, challenging the view that the brain alone generates mental states[42]. Rather than opposing such perspectives, a causal BPS framework can treat the brain

as one dynamically embedded node within a larger system. Neuroimaging and interventional studies, including sham-controlled transcranial magnetic stimulation trials, show that modulation of neural circuits can alter mood[43–45], while environmental and relational changes likewise reshape neural function[46,47].

The implication is not that one level holds causal priority, but that causal influence is distributed and reciprocal across levels. The clinical task is therefore not to privilege the brain, nor to sideline it, but to understand how neural, psychological, and social processes co-constitute one another over time. The limited translational yield of reductionist biological psychiatry over past decades underscores the need for integration rather than abandoning biology.

## Application for causal integration during the next fifty years

Fifty years after Engel's original call, the task is no longer to defend the BPS model against biological or psychosocial dogmatism but to rebuild it scientifically by uniting medicine, psychology, neuroscience, public health, and social sciences through cross-level causal modeling in research and training. Without such cross-disciplinary adoption, the BPS framework risks remaining aspirational. As psychiatry becomes increasingly measurable, philosophy can illuminate conceptual assumptions, clarify explanatory levels, and refine definitions, without substituting for empirical inquiry. Its value lies in clarifying reasoning without competing with science. Such discipline is essential if psychiatry is to remain aligned with medicine[48] and keep the BPS model teachable and applicable through shared empirical reasoning.

The first step is to make the BPS model causal. Causal inquiry must extend upward and downward across levels, from molecular dynamics to relationships and cultural context, without reducing one to the other.

Modern analytic tools such as directed acyclic graphs, structural equation modeling, and computational psychiatry can specify pathways across levels of organization[49]. Predictive coding and active inference frameworks formalize how the brain generates predictions, propagates prediction errors, and engages in action to minimize them across brain-body-environment dynamics[50], yielding testable hypotheses for information flow across systems[51]. However, while productive, predictive processing accounts are mechanistically underspecified, raise falsifiability concerns, and risk treating brain-centric metaphors as resolved, particularly regarding social and structural levels of explanation[50]. Network approaches and graph-theoretical modeling further allow estimation of dynamic relationships among symptoms, behaviors, and biological variables over time, capturing how feedback loops stabilize or destabilize health states[52].

The second step is to make the model measurable. Digital phenotyping, wearable sensors, and ecological momentary assessment now allow physiological, psychological, and social variables to be tracked with the same temporal precision as laboratory assays[53]. Time-resolved neurophysiological and autonomic data reveal how relational stress alters neural synchrony[47] and how neural states predict social withdrawal[54]. Emerging biological pathways and methodological approaches that operationalize this causal framework are summarized in Table 1.

Third, the model must become personalized. Disorders such as depression comprise multiple mechanistic subtypes from inflammatory to trauma-related to circadian that require distinct treatments, from anti-inflammatory agents to psychotherapy or chronotherapy[55]. Incorporating precision medicine, machine learning can integrate multimodal data to identify such subtypes and match interventions accordingly, leading to targeted and individualized treatment plans. A refined psychiatric nosology will be essential, since current diagnostic categories are too coarse to align with mechanisms[56].

**Table 1 | Emerging pathways and methods for testing a causal biopsychosocial model**

| Level | Example concept | Empirical anchor/measurable link | Illustrative relevance to the biopsychosocial framework |
|---|---|---|---|
| Molecular/cellular | Epigenetic regulation | Changes in deoxyribonucleic acid (DNA) methylation and histone acetylation in response to stress or psychotherapy | Shows how psychosocial inputs leave stable biological marks |
| Microbiome-immune axis | Gut-brain-microbiota interactions | Cytokine modulation, tryptophan-kynurenine pathway | Links diet and stress with mood via immune signaling |
| Cellular energetics | Mitochondrial function and oxidative stress | Adenosine triphosphate (ATP) production, reactive oxygen species (ROS), and mitochondrial DNA (mtDNA) copy number | Connects fatigue and stress resilience to metabolic regulation |
| Chronobiology | Cellular clocks and circadian misalignment | Clock gene expression, melatonin and cortisol rhythms | Relates social schedules and sleep disruption to inflammation and mood |
| Neural network integration | Functional and structural connectivity | Resting-state functional magnetic resonance imaging (fMRI), electroencephalography (EEG) coherence, and neuromodulation effects | Maps how psychological states alter neural communication and self-regulation |
| Neurovisceral integration | Anti-inflammatory vagal reflex | Heart rate variability (HRV) indices, vagal tone, and cytokine suppression | Demonstrates direct pathways linking emotion and immune modulation |
| Systemic network level | Predictive coding and active inference | Multilevel modeling of brain–body feedback | Provides a framework for estimating expectation effects on physiology |
| Endocrine–metabolic coupling | Hypothalamic–pituitary–adrenal (HPA) axis and glucose–lipid regulation | Cortisol profiles, insulin sensitivity, inflammatory mediators | Illustrates chronic stress effects bridging mental and physical health |
| Causal inference methods | Randomized controlled trials, quasi-experiments, Mendelian randomization, natural experiments | Experimental and quasi-experimental causal models distinguishing correlation from causation | Enable behavioral and biological mechanisms to be tested rather than merely described; identify mediators linking psychosocial exposures with biological outcomes |
| Behavioral/social level | Digital phenotyping and ecological data | Smartphone sensors, voice tone, sleep metrics | Captures real-time psychosocial–biological coupling in daily life |
| Structural social level | Socioeconomic gradients, discrimination, neighborhood deprivation | Education, income, area-level deprivation scores | Upstream drivers of chronic stress, psychological distress and biological embedding |
| Macrosocial/environmental level | Resource inequity, urban stressors | Geospatial metrics for healthcare utilization, food deserts, noise metrics | Upstream macrosocial initiation of cross-level biopsychosocial interactions |

**Table 2 | Vision for 2077: Causal biopsychosocial integration realized**

| Vignette (clinical scenario) | What is done by the interdisciplinary team | Best outcome (biopsychosocial integration realized) |
|---|---|---|
| **Gastroenterology**<br>A man with inflammatory bowel disease works in a high-strain job with unpredictable shifts. He wears a biosensor tracking inflammation and sleep. Rising cytokines and disturbed sleep signal an impeding flare. | The care team receives an automated alert. The gastroenterologist adjusts biologic therapy, while the psychologist addresses distress and coping, and the social worker negotiates flexible scheduling and job protections with the employer to reduce job strain. Rather than treating inflammation and distress as parallel targets, the team intervenes on the occupational driver sustaining both. | Working synergistically, the team improves treatment adherence, reduces work-related strain, and stabilizes circadian rhythms, illustrating how modifying the workplace closed the causal loop across biological, psychological, and social systems. |
| **Psychiatry**<br>A young woman with recurrent depression in remission lives in unstable housing and works a precarious gig job with irregular hours, while caring for a younger sibling in a context of ongoing family conflicts. She uses an app integrating voice tone, sleep, activity, and self-reported experiences of workplace harassment and social isolation. Predictive algorithms, combining these social and behavioral data with inflammatory and neural markers, forecast relapse. | The system flags risk to her clinician. The care team initiates targeted interventions, including transcranial magnetic stimulation or pharmacotherapy where indicated, psychotherapy focused on role transitions, family-based interventions, and social measures such as advocacy for safer working conditions and linkage to housing support. Instead of focusing solely on symptom recurrence, interventions target the social conditions and relational stressors forecasting relapse. | By jointly addressing housing insecurity, work exploitation, and family dynamics, emotional stability and daily functioning are maintained and hospitalization is prevented |
| **Cardiology**<br>A male patient in his sixties with coronary heart disease lives in a segregated, high-crime neighborhood and faces chronic job insecurity. His smartwatch detects repeated fear and anger episodes during long commutes and hostile workplace interactions. Causal modeling links these context-bound stress-related physiological surges with coronary plaque activity on cardiac imaging before coronary crises occurs. | Integrated dashboards unify laboratory results, digital biomarkers, psychosocial metrics, and neighborhood data, allowing the care team to intervene proactively. Alongside optimizing statin and antihypertensive therapy, they deliver brief stress-reduction and anger regulation training, and collaborate with occupational health and community services to modify work demands and explore safer commuting or partial remote work. Rather than adding stress management to optimized medication, the model identifies context-bound physiological surges as modifiable upstream triggers. | By changing upstream stressors and environmental exposures, not only individual coping, plaque activity and cardiovascular risk decline, and emotional regulation improves, exemplifying preventive biopsychosocial medicine that reduces stress-related triggering of a heart attack. |

Finally, the model must be embedded in systems and structures. Engel's vision encompassed healthcare delivery and individual care. Team-based integration of medical, psychological, and social expertise provides biopsychosocial collaborative care, which improves outcomes[57], but remains underused and underfunded[58]. Beyond this implementation gap, a future framework must address structural determinants, including poverty, discrimination, climate stress, forced migration, transportation, and sanitation[59,60]. Without this macro-level integration, BPS medicine risks devolving into micro-adjustments that leave upstream causes untouched.

A causal understanding does not automatically translate into better care; implementation requires translation into decision-making algorithms, training, and resource allocation. However, a causally informed framework can improve practice by distinguishing primary drivers from downstream correlates and mediators within a given case. This enables prioritization of upstream biological, psychological, and social determinants, sequencing of interventions according to modeled interdependencies, and avoidance of redundant or low-yield treatments. Rather than adding biological, psychological, and social measures in parallel, the aim is to intervene where cross-level effects are most likely to propagate. These principles are illustrated in Table 2, which envisions how causal biopsychosocial integration could transform everyday clinical practice by 2077.

## Concluding remarks

Fifty years after Engel's call, medicine faces a choice. Psychiatry, frustrated by biological false starts, risks abandoning the BPS model; medicine, secure in its biomedical anchors, risks ignoring it. Both are mistakes. Allied disciplines should adopt cross-level causal modeling through training and research. Engel's lasting contribution was not eclecticism but the call for multilevel causation as scientific method. His critics err when they elevate one domain above the others.

The task now is to transform the BPS model from metaphor into a causal, measurable, and systemic science of interactions. Over the next fifty years, a mature BPS science may render rigid separations between biological and psychosocial categories increasingly obsolete. By 2077, the only meaningful question will be where in the system to intervene so that the effects converge, altering disease course and illness experience, to truly restore health. Achieving this vision requires investment in interdisciplinary training, data integration, implementation science, and policy commitment to structural interventions.

## Data availability

All data are available in the main text. No new data, code, or materials were generated for this work.

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

## Acknowledgements
The author acknowledges the use of ChatGPT (OpenAI, GPT-5) to assist in language refinement and stylistic editing during manuscript preparation. The content, structure, and interpretation of all scientific arguments are solely the author's own. This work received no specific grant from any funding agency in the public, commercial, or not-for-profit sectors.

## Author contributions
Conceptualization, writing of original draft, review & editing: RvK.

## Competing interests
The author declares no competing interests.
