## [Transparent Peer Review file · Communications Medicine]

Applying a causal biopsychosocial model to guide medicine and psychiatry fifty years after Engel

Corresponding Author: Professor Roland von Känel

Version 0:

Reviewer comments:

Reviewer #1

(Remarks to the Author)

Thank you for the opportunity to review this manuscript. I found the author's commentary to be insightful and agreeable, and a fitting way to mark the golden anniversary of Engel's biopsychosocial model. Altogether, I believe the manuscript makes a worthwhile and timely contribution to the literature, and I am happy to recommend it for publication in your outlet.

I do have a number of more specific comments, which I have raised below both for the editor's and author's consideration:

1. On p. 1, lines 38-41, the author claims: "In this framework, "causal" is understood in the sense of regulatory interactions between domains, in contrast to causality defined by probabilistic dependencies among variables across levels, which avoids implying determinism."

Personally, I do not see the need for this distinction, and therefore feel that the author can delete this sentence altogether. Indeed, in some cases, there will in fact be probabilistic dependencies among variables; in such cases, the challenge should be in differentiating such causes from those belonging to regulatory interactions. This speaks to the complexity of biopsychosocial modelling, not to mention analysing aetiology more broadly. As such, I do not believe we should shy away from this challenge.

2. In the section titled "Achievements", the author has provided a well picked collection of examples to demonstrate his case. However, the author fails to distinguish between cases where the BPS model has explicitly motivated these research avenues, or instead, can simply be applied post hoc in order to explain relevant findings. More generally, I felt that the author could have elaborated on how the biopsychosocial model might be translated into a clear, multilevel research heuristic to guide hypothesis generation (although to be fair, this is touched on in the penultimate section).

3. On p.2, lines 51-58, the author states: "As an example, patients with inflammatory bowel disease show a 1.5-fold increase in the prevalence of depressive symptoms during flare-ups, mediated by proinflammatory cytokines that disrupt mood and sleep. Likewise, up to 40% of cancer patients treated with interferon- α develop major depression. Here, biology initiates the cascade, where immune activation induces psychological change, which then feeds back into disease course. Conversely, psychosocial interventions can enhance immune function and dampen inflammation, underscoring the bidirectional nature of these effects."

As an aside, the causal arrow can go both ways of course – there was considerable debate about a decade ago, with some arguing that depression was caused by inflammation, and others suggesting the opposite was true. There is now a convincing body of work to suggest that inflammation should not be attributed causal primacy for depressive illness, although its importance as an aetiological mediator is widely recognised (e.g., between social stress and depressive onset). For a discussion, I would suggest the author considers the following:

Berk, M., Williams, L. J., Jacka, F. N., O'Neil, A., Pasco, J. A., Moylan, S., ... & Maes, M. (2013). So depression is an inflammatory disease, but where does the inflammation come from? *BMC medicine*, 11(1), 200.

Slavich, G. M., & Irwin, M. R. (2014). From stress to inflammation and major depressive disorder: a social signal transduction theory of depression. *Psychological Bulletin*, 140(3), 774.

Watt, D. F. (2023). The separation distress hypothesis of depression—an update and systematic review. *Neuropsychanalysis*, 25(2), 103-159.

In any event, I think it is worth noting that social precipitants and psychological states, such as depression, might themselves also initiate inflammatory responses. If anything, this underscores the author's own point.

4. I was impressed by the author's section on "Critiques", which, in my opinion, was incisive, succinct, and largely on the mark. However, I did wonder how realistic it is for the author to expect researchers to "rigorously define and specify causal pathways" (p.3, lines 92-93), particularly given the slow progress made in clinical neuroscience, machine learning, computational psychiatry and psychiatric genetics (just to cite a few examples) to clearly and definitively identify the candidate mechanisms responsible for causing mental disorder. In my own field, for example, the more we learn about the various mechanisms that underlie depression, and the complex, bidirectional interactions between them, the more difficult it has become to specify distinct causal pathways (indeed, the abovementioned debates surrounding inflammation is a clear case in point). Given such complexities, I would be interested to learn how the author thinks researchers might reliably navigate such complexities in order to rigorously define causal pathways in the first place. (That being said, I also recognise that this very point probably deserves a full paper in its own right, so I'm not sure that the author's current submission is the right place to address this).

5. In closing, I would suggest that the author include graph theory or network analysis as a promising modelling approach to help us capture biopsychosocial dynamics; particularly longitudinal approaches, which allow us to investigate how interactions between precipitating and perpetuating factors across different levels evolve over time. See, for example: Borsboom, D., & Cramer, A. O. (2013). Network analysis: an integrative approach to the structure of psychopathology. *Annual Review of Clinical Psychology*, 9(1), 91-121.

I hope the author finds my comments helpful, and I would like to take this opportunity to wish him all the best in his future endeavours.

Reviewer #2

(Remarks to the Author)

This is a timely and well-constructed position paper that brings together recent research to defend the ongoing relevance of the biopsychosocial (BPS) model and articulate its unrealized scientific potential. The authors' call for "a causal biopsychosocial framework, an integrative, measurable and predictive science of interacting systems uniting medicine and psychiatry, to guide the next fifty years of research and care" resonates with many in the field, and while not novel in its central claims, the paper makes a valuable contribution by consolidating and reviewing with detail recent advances in defense of a position that diverges meaningfully from some recent engagements with the BPS. The question of how to get there, however, remains, and it is here that the paper would benefit from deeper engagement with existing integrative work and from greater precision in some of its formulations.

The opening observation that Engel's model "has often become a checklist rather than a causal framework" is well-taken, but raises a question the paper does not fully address: why has this been the case? Part of the answer likely lies in the very problem of integration the authors invoke. One of the most persistent critiques of the BPS model concerns the limited articulation of how biological, psychological, and social processes actually interact — and the authors do not engage substantively with this critique, nor with recent attempts to advance it, including our own work. This is a notable gap given the paper's ambitions and synergies with this work. The cultural-ecosocial systems (CES) approach (Gómez-Carrillo & Kirmayer, 2023; Gómez-Carrillo et al., 2023; Kirmayer, 2024) represents precisely the kind of recent, rigorous attempt to reframe the bio/psycho/social — for which Engel provided little in terms of integration — as multi-level interacting processes with feedback loops. The CES account aligns closely with the direction the authors advocate, and additionally foregrounds culture, subjectivity, and narrativity as constitutive parts of mechanism rather than contextual add-ons. Whether or not the authors choose to engage with CES specifically, any integrative account of the BPS that aspires to the authors' stated goals will need to grapple with these dimensions.

This brings us to what is perhaps the most important conceptual tension in the paper. The authors mention the role of "perceived" stress without fully reckoning with what taking perception and meaning seriously implies for the framework. The paper later asserts that philosophy's "value lies in clarifying reasoning — distinguishing mechanisms from meaning — without competing with science," but the distinction between mechanism and meaning as drawn here is not straightforward and risks reproducing a separation that much recent work has sought to overcome. We and others have argued that meaning is not separate from or epiphenomenal to biological mechanism but is intrinsic to it — and that this is precisely one productive avenue for advancing integration across the biological, psychological, and social. Kirmayer has shown that cultural meaning and narrative actively shape symptom formation and healing rather than merely interpreting them. Varela, Thompson, and Rosch's *The Embodied Mind* (1991) grounds this in enactivism, arguing that cognition is fundamentally sense-making and that meaning-generation is the basic activity of living systems. Fuchs extends this by showing how meaning becomes sedimented in bodily habit and affect, while Deacon's *Incomplete Nature* (2012) makes perhaps the most ambitious version of the claim — that semiotic and intentional processes are emergent features of physical dynamics,

neither reducible to mechanism nor separable from it. Taken together, these accounts challenge the assumption that biological mechanism is a meaning-free substrate onto which significance is subsequently projected. The paper moves in this direction in places, but then appears to prioritize approaches to integration that risk reductionism and leave insufficient room for subjectivity, meaning, and the "perceived" — perhaps inadvertently so.

Several specific formulations would benefit from revision. The abstract claim that contemporary tools "now make it possible to test and quantify these interactions across levels of organization" is stronger than the evidence warrants; these tools are better described as enabling simulation and, in some cases, testing of such interactions, with quantification remaining partial and context-dependent. Similarly, the claim that advances in psychoneuroimmunology, social neuroscience, and computational psychiatry "map the causal architecture" of the BPS overstates current achievements; a more precise formulation would be that they are beginning to articulate specific models of integration across levels, in some cases with claims bearing on causality.

The paper's reliance on predictive processing accounts is productive but would benefit from acknowledgment of their limitations — including mechanistic underspecification, concerns about falsifiability, and the risk that what remains largely brain-metaphorical is treated as mechanistically resolved, particularly with respect to social and structural levels of explanation.

The claim in the table that multilevel modeling "quantifies top-down expectation effects on physiology" is similarly overstated; a more defensible formulation would be that it "provides a framework for estimating expectation effects on physiology."

The paper's treatment of critiques also warrants care. Kandel is cited as representative of the position that "all mental processes are ultimately biological, rendering psychosocial explanations derivative," but this characterization is reductive. Kandel does prioritize biology as the foundational explanatory level, but his argument — that psychosocial interventions are real and powerful precisely because they produce measurable changes in brain structure and gene expression — makes him better described as a biologizing integrationist than a dismissive reductionist.

Likewise, the characterization of Fuchs as inverting the biological hierarchy, "claiming relationships and embodied experience, not biology, should anchor explanation," does not do justice to the subtlety of his argument. Fuchs is not simply flipping the hierarchy; he is challenging the assumption that biological processes constitute the primary or self-sufficient explanatory level, arguing instead that embodied experience and relational context are constitutive of, not merely dependent on, the biological. The distinction matters, and collapsing it misrepresents what phenomenological approaches actually claim.

A similar concern applies to the paper's characterization of enactivist perspectives as proposing that "the brain is a mediating organ, merely mirroring lived experience rather than actively generating it." This formulation inadvertently reproduces the very dualism enactivism seeks to overcome. Enactivism does not demote the brain to a passive mirror — it challenges the assumption that the brain alone generates experience from the inside out. The brain remains an active participant, but one node within a larger dynamic system involving body, environment, and relational context; activity and agency are distributed, not eliminated. De Haan's enactive approach to BPS interaction is specifically concerned with reconceptualizing how biological, psychological, and social dimensions are structurally intertwined, rather than ranking them or assigning passive versus active roles. At minimum, the word "merely" should be removed, and the authors may wish to revisit the characterization more substantially.

The concern that "without causal modeling, the BPS approach devolves into checklists and loses explanatory power" is understandable in intent, but risks being reductionist in effect — not because causal rigor is unimportant, but because privileging causal modeling as the criterion for explanatory power implicitly favors the biological level, where such modeling is most developed, and forecloses interpretive, narrative, and phenomenological forms of explanation that may be more appropriate at psychological and social levels. The call for theoretical rigor and integrative accounts is well-founded; the anchoring of that rigor specifically in causal modeling may inadvertently narrow the framework it seeks to renew.

These concerns notwithstanding, the paper contains genuinely strong passages. The observation that "functional and mental disorders are not failures of the BPS framework but laboratories for its evolution" is elegantly put and captures something important. With revisions as suggested above and greater engagement with existing integrative literature, this paper could make a more substantial contribution to what it aims to advance.

Reviewer #3

(Remarks to the Author)

In this Perspective, the author (re)builds a case for proper adoption of Engel's biopsychosocial model (BPS) in medicine. The author covers the BPS' history, its achievements and its critics, after which they turn to what we can still gain by proper adoption and implementation of Engel's recommendations.

Before I continue this review, I want to stress two elements of my background that are relevant for my review. First, my areas of expertise are philosophy of psychology/psychiatry, psychiatric ontology and causal modelling, due to which I cannot evaluate the accuracy of claims about medicine (or the adoption of BPS) beyond those falling roughly within the purview of psychiatry and neuroscience. I therefore have not covered those in this review, but my silence on these topics should not count towards any evaluation of their quality; I simply cannot assess this. Second, I am broadly aligned with general systems

theories, due to which I may be overly receptive to the claims presented here relative to the overall intended audience or other experts in my field. My review therefore may be quite friendly to the ideas presented here, and in absence of highly critical reviews I would recommend the author or editor to perhaps search for further (informal) review.

This being said, my overall opinion of the article is quite positive. Beyond my general agreement with its claims, it is well-written and its message is clear, although some turns of phrase in my view could be changed for accuracy or removed (more on which below). The position taken and described here is – obviously – not new, being a call for adopting a previously introduced model, but the author in my view convincingly argues his point whilst also adequately portraying critiques. Beyond this, I am also of the opinion that the message of the article is important, timely and – like Engel's original article – potentially influential.

Nevertheless, I feel the article has some remaining general weaknesses, and specific elements that could be improved. For the author's ease, I've numbered the former, and alphabetized the latter below.

1. From the biomedical to the biopsychological model?

Throughout the article many examples are used to exemplify the BPS, but few if any of these are examples of the social factors within the biopsychosocial model. This is reinforced by some factors that are presented as 'psychosocial', but which nonetheless refer to internal, psychological processes (e.g., emotion, subjective experience, expectations). In turn, it often seems like the BPS consists of two levels; the biological and the psychosocial, as opposed to the social being a standalone level.

To be specific, this applies to all examples under the paragraph 'Achievements' except for 'the clinician's words', and all examples used as counter-arguments in 'Lessons'. In the paragraph 'The Next Fifty Years', social factors are only mentioned in two sentences at the end, from "Beyond this implementation gap..." through "...leave upstream causes untouched.". Nearly all discussion of mechanism and measurement also only involves biological and psychological variables here. In Table 2, only one social factor (i.e., work-related strain) and related intervention comes up as well (i.e., workplace intervention).

I am not sure whether this is the author's intention; it seems unlikely given the inclusion of social factors more generally. Even so, I would argue that it was Engel's intention to not conflate social factors with psychological ones; social factors can directly affect biological and psychological factors independently, and without mediation by the other (e.g., poverty -> diet -> obesity/diabetes even if the person might want to eat healthy).

I don't know enough about medicine to provide good examples of existing attention to social factors; preventive medicine comes to mind (e.g., condom availability interventions), but is probably independent from the BPS model. Yet, in Table 2, the cardiology example might be enriched by investigating external causes of anger (e.g., job, family, personality, living situation) prior to committing to anger management therapy, which would pay attention to social factors instead of only psychological factors. Similarly, the psychiatric example could be enriched with attention to the young woman's living circumstances that are causally relevant to the depressive state (e.g., problematic family, school or work situations like bullying; sleep interruptions). Ensuring that social and psychological factors are both separately included is useful elsewhere too.

2. Adoption of the BPS is a multi-disciplinary endeavor

The author's hopes for the next 50 years require a causal integration across the different levels of the biopsychosocial model. I wholly agree with the author on the need for such integration, but I think this requires adoption of the BPS model and program beyond medicine to be successful. This is tacit in the current article – unless medical researchers are to create causal models of how psychological and social factors interact – but should probably be made explicit.

To support this; psychologists and neuroscientists for example pay very little attention to social factors. This is not just the case in clinical psychology, but even in social and industrial-organizational psychology. Interventions mostly focus on intervening on internal variables like attitudes, beliefs or motivations, as opposed to changing structural environmental or social factors, if these are even considered.

In short; interdisciplinary interventions require all of these disciplines to be on board, and thus would likely also require changes in training in those other disciplines. Mentioning that bringing these other disciplines on board is also a relevant task here would be useful, to prevent disappointments or failure. I do not think this requires novel paragraphs, but could be mentioned in both the 'Next 50 Years' and 'Conclusion' sections.

A. Line 61-64: "A longitudinal study..." to "...translates into biological risk."

In these lines the author uses a so-called 'weasel word' (i.e., X translates into Y) to describe a predictive relationship between subjective experience and biological processes. This creates unnecessary confusion about the relationship, since 'translate' can also be applied to causal relationships (e.g., 'the amount of alcohol sold translates into vehicular manslaughter cases'). I would advise a different formulation here, like 'how subjective experience can be relevant for estimating biological risk'.

B. Line 101-103: "Far from exposing the model's limits, such discoveries show its refinement: functional and mental

disorders are not failures of the BPS framework but laboratories for its evolution.”

I think the author here wants to say that the ongoing study of mental and functional disorders shows the relevance of the BPS model (i.e., the interdependence and intertwinedness of bio-, psycho-, and social mechanisms relevant to disease, illness and sickness). The current sentence however does not show that clearly: I'm not sure what the 'limits' are that are referred to, nor how functional and mental disorders even could be failures of the BPS as opposed to 'failures' of a nervous, mental and/or social system. To state that any disorder should be considered a laboratory for the evolution of a model may be a nice turn of phrase, but it also sounds awfully unethical when those disorders can come with massive suffering.

In other words, I don't think this sentence expresses what the author wants, and could lead to negative reactions (i.e., are people with mental disorders 'laboratories for the evolution of the BPS'?). A different formulation might be preferable to make the point of the author come across better.

C. Line 110-120: “The “brain as secondary” error...” to “...within the BPS network.”

From a philosophical perspective, several things go wrong in these lines.

First, the author introduces a 'brain as secondary'-error, which he contrasts with neuroscientific evidence that the brain can be intervened upon to change experience. Specifically, the former error is to say that the brain is a mediating organ for lived experience, not “a generator of experience”. Note here that the possibility of intervention on experience through the brain is independent of its status as such a 'generator', especially given that the ones argued to commit the error are enactivists like De Haan.

The evidence the author cites after all is not necessarily relevant to the enactivist point. Enactivists could indeed argue that the function of the brain is to be this mediating organ, which facilitates engagement with the world. But I doubt they would claim that experience only depends on the environment or relationships. As far as I know, enactivists more generally merely argue that experience is not 'generated' by the brain in the sense that experiences of external objects and their properties are merely brain-generated 'illusions' or 'representations' derived from passively received input. Instead, they argue that experience/cognition also already implies and directly follows from action, through sensorimotor loops (i.e., loops between sensory brain areas and motor brain areas due to co-occurrence, both internally through Hebbian learning and externally through co-occurrence - movement causes sensory changes, and sensing differences identifies action opportunities that also 'prime' future sensory changes). Although direct intervention on the brain can induce certain experiences, these are generally different in quality than normal experience due to this absence of other elements of the sensorimotor loop (e.g., compare seeing an elephant to imagining one; compare noticing the effects of an adrenaline shot to being angry or afraid). It is in this sense that they would argue that the brain is not a pure 'generator of experience', even though you can artificially induce experience through manipulating it directly. In turn, being able to predict emotions from neurological patterns is not necessarily at odds with enactivism, nor the possibility of intervening on mood by manipulating the brain. Experience is created through a continuous loop between the organism (including the brain!) and the environment – and we can predict the next step in that loop through looking closer at any element of it. After all, I can also predict emotions from behavior (e.g., crying predicts that someone is sad internally) and brain input caused by others (e.g., hitting someone predicts they will get angry, afraid or sad).

Second, the author shows that 'targeted modulation of neural circuits can alter mood when environment and relationships remain constant', and suggests this means that the brain is not a passive recipient but an active node in the BPS network. Yet, this argument is logically unsound, independent of whether it targets enactivism correctly. This is clear if we use a different example: if I were to argue that diet determines what is digested, one could also counter that interventions on the esophagus (e.g., blockage) can directly alter what is digested even if diet remains constant. Yet, this was never excluded by the initial argument. Key here is that the first argument is about normal circumstances (e.g., mediating variables are in a 'normal' state), under which the causal claim holds. The same could apply to the enactivist claim.

Overall, I therefore think this section needs to be rewritten. The current arguments in my view miss the mark, but also are not entirely necessary. To make the point the author intends to make, it is sufficient to show that we should not privilege one level over another needlessly, which need not involve attacking the foundations of philosophical positions. Proper engagement with any such philosophical positions also would require far more words, without necessarily making the point clearer. Lastly, even if the enactivists are right about everything, it would still be useful for them to consider biological interventions as well – it is enough to state that to make a case for the BPS.

In the context of medicine, the fact that the brain is relevant could for example be simply supported with neurological deficits affecting cognition (e.g., Alzheimer's, Wernicke's aphasia, Korsakoff). Even if the brain merely mediated everything, it is still medically relevant as this mediator. It is also possible to directly intervene on it to bring about effects. It is unnecessary to bring in philosophical tarpits like causal priority, free will, intention, or consciousness here to indicate that we should always pay attention to the different levels and/or their interrelationships.

To put it very bluntly; the BPS model stands to the biomedical model within the health sciences, as does the philosophy of a racing team to that of a car mechanic. If a car deviates from a preset 'norm' according to its driver (illness), the mechanic looks at the car's inner machinery (biological). A racing team instead also looks at the driver (psychological) and the track + other competitors (structural and social factors)). The driver might complain mainly because they lack skill/focus (psychological), or because the track doesn't suit them (structural), or because competitors are better/outperforming them (social factor). And perhaps something is wrong with the car (biological). Changing something about the car could improve

the situation only relative to these other elements as well, as well as the goal (e.g., more top speed is only good if this is useful on the track, and the driver can handle it, and if this is where we are falling behind or could overtake). Meanwhile, looking only at the track, the competitors, the driver or the car is not a good idea for a racing team; if the car isn't any good, we can change tracks and competitors as much as we can, but it won't help. Bad drivers also can't make a good car run.

This metaphor is not entirely apt, as it ontologically separates the factors too much (please do not use it!). The point of it however is that showcasing interdependence is sufficient to claim that the BPS outperforms the biomedical model or 'brain as secondary' approaches: it is stupid to ignore elements that we could use or that in fact do matter for outcomes. This line of reasoning is also present in the conclusion (i.e., 'The only meaningful question will be where in the system to intervene so that the effects converge...'). Perhaps an argument along such a line, as a contrast to focusing on a subset of the system, would serve the purpose of this section better than the current one. Within such arguments, positions like De Haan's (citation 33) could also be considered allies to the BPS (e.g., introducing 'existential' factors to the model!), even if more stress on the biological is desirable to the author.

D. Line 120-129: "The problem of intention..." to "...override biological predispositions."

I would advise the author not to make use of Libet's experiments as an example, and also not to make claims about the reducibility of intentions. Both topics are highly controversial in philosophy, and have their own attached secondary literatures, as the text already shows due to the included nuance. In the current formulation, it is also not clear how/why we should conclude that psychological phenomena are not reducible to biological processes on the information given; it seems we should partially trust on conscious veto or contextual modulation, but the former is Libet's controversial interpretation, whilst the latter seems to potentially support the enactivists' point. Supporting the conclusion would require more supporting argumentation or explanation of the author's reasoning.

I think my advice under C applies here too; it is not necessary to invoke philosophically charged debates here. Instead, I would once more advise to draw attention to the importance of the interrelationships between the different levels.

E. Line 138-139: "Its value lies in clarifying reasoning – distinguishing mechanisms from meaning – without competing with science."

Although I overall agree with the author on the value of philosophy, I am not sure what is meant by 'distinguishing mechanisms from meaning' here. Upon consideration, this vaguely brings philosophical clarifications like not reifying depression into mind (i.e., depression refers to a cluster of symptoms, not the cause of these symptoms), but I am not sure whether that is what is meant here.

I therefore would leave out this part; it is not crucial for understanding the sentence.

Reviewer #4

(Remarks to the Author)

This is a fiery paper that joins a recent chorus of critiques of the biopsychosocial model from the viewpoint of causation. The paper hits at the weakest spot in the biopsychosocial model while nevertheless recognizing its importance and the fact that it cannot be easily jettisoned. It is therefore a welcome addition to this growing literature.

Still, there are a few issues that I'd invite the author to address.

First of all, as I mentioned, the author is not the first to propose a causal refurbishing of the biopsychosocial model. The piece sits in the wake of quite a few thinkers within the active inference paradigm (some of whom are not mentioned: e.g. Van den Bergh et al., 2017), enactivism (e.g. Fuchs, 2017; De Haan, 2020, who are mentioned but quickly dismissed) or philosophical approaches such as Ongaro's (2024) who have recently synthesised the two while pushing for the causal understanding of the model. Positioning the article as part of a broader series of critiques would be far better than brushing aside any other position that have argued pretty much the same thing, namely that the biopsychosocial model needs to be rethought in a causal and integrative way.

The other problem I see in the paper is the under-analyzed gap that exists between reaching a causal understanding of the model and bearing its fruits in clinical practice, which is the ultimate goal of the model. Just exactly how does the author think that a causal integration between bio, psycho, and social factors can lead to better clinical practice and outcomes? I'm not saying that it doesn't, but the link should be discussed. Let's take his example of the oncology patient with tumor, distress, and marital discord. Let's assume that the score of scientific advances the author mentions in the article (e.g. neuroimaging, digital phenotyping, directed acyclic graphs, machine learning for precision medicine, etc) allows us to understand how these bio, psycho, and social factors causally interact with one another. What's next for our patient? What can a causation-informed approach do for that patient that the checkered approach—which might try cancer therapy, psychotherapy and forms of social therapy all at once without understanding precisely how these causally interact—does not? (see Mescouto et al., 2020; Ongaro 2024 for similar concerns). There is a yawning gap between reaching a scientifically advanced picture of biopsychosocial integration and having therapeutic effects. Unless the author acknowledges that there is such gap, his polemic is built on sand.

Towards the end of the article, the author gestures at the 'structural determinants' like poverty, discrimination, sanitation etc. that—hard data at hand—massively impact psychiatric health. Yet it is obvious that to achieve change in this direction it's political change that we need, first and foremost, more than scientific understanding of causal integration. The author is right

in stating that “biopsychosocial collaborative care, team-based integration of medical psychological and social expertise improves outcomes but remains underused and underfunded.” Clearly, it’s the underfunding of social measures that most cripples the aspirations of the biopsychosocial model.

A couple of minor passages that invite clarification:

Line 113: Relational and enactivist approaches “remain theory-driven phenomenological interpretations without mechanistic proof that person-world dynamics function as causal determinants of experience.” The authors in question would reply that enactivism serves as a framework for interpreting these mechanisms (which, by the way, they partly discuss; see Fuchs’ work). Predictive processing is another similar framework.

Line 137 “As psychiatry becomes increasingly measurable, philosophy should illuminate concepts, not replace evidence or fill empirical gaps with metaphor. Its value lies in clarifying reasoning—distinguishing mechanisms from meaning—without competing with science.” Can the author give precise examples of how philosophy of psychiatry has replaced evidence, filled gaps with metaphor, and competed with science? Also, what does “distinguishing mechanisms from meaning” actually mean?

Despite these issues, I believe the paper jolts the discussion in a positive direction and brings up to date all scientific advances that will provide a much sounder picture of how biological, psychological, and social factors causally interact—if not in the near future, then likely within the next 50 years.

References cited:

- De Haan, S. (2020). *Enactive Psychiatry*. Cambridge University Press.
- Fuchs, T. (2017). *Ecology of the Brain: The phenomenology and biology of the embodied mind* (Illustrated edition). OUP Oxford.
- Mescouto, K., Olson, R. E., Hodges, P. W., & Setchell, J. (2020). A critical review of the biopsychosocial model of low back pain care: Time for a new approach? *Disability and Rehabilitation*, 1–15.
- Ongaro, G. (2024). Outline for an Externalist Psychiatry (1): Or, How to Fully Realize the Biopsychosocial Model. *Philosophy, Psychiatry, & Psychology*, 31(3), 269–284.
- Van den Bergh, O., Witthöft, M., Petersen, S., & Brown, R. J. (2017). Symptoms and the body: Taking the inferential leap. *Neuroscience and Biobehavioral Reviews*, 74(Pt A), 185–203.

Version 1:

Reviewer comments:

Reviewer #1

(Remarks to the Author)

The author has done a fine job addressing the wide range of reviewers’ concerns, and I am happy to recommend their revised manuscript for publication. The only suggestion I would have is to retain the start of their title - "Fifty years after Engel" - which has a little more impact than their revised one.

Reviewer #4

(Remarks to the Author)

I believe the author has successfully addressed the point I raised. They have now situated the manuscript within broader causal reformulations of the BPS model by engaging with other frameworks that advance a similar causal perspective (active inference, enactivism, externalist psychiatry, etc.), and they have also clarified how causal modeling can yield concrete clinical benefits (which, to me, still seems somewhat optimistic, but we will have to see). Overall, they have made the piece more nuanced and better positioned within the broader discussion of the BPS model. I think it is now in publishable form.

Responses to Reviewer Comments – Manuscript/Submission ID: COMMSMED-25-2910: “Fifty Years After Engel: The Biopsychosocial Model Needs to Be Rebuilt as a Causal Science”

Now entitled “Applying a causal biopsychosocial model to guide medical and psychiatric research and practice”

Comments from Reviewer #1

Comment 1 – Regulatory vs probabilistic causality distinction

The reviewer suggested deleting the sentence distinguishing “regulatory” from “probabilistic” causality.

Response

I agree that this distinction was unnecessary and potentially confusing. The sentence has been deleted and the causal framing simplified.

Location of change: Introduction section “*From Blueprint to Rhetoric: Back to Science*”, top of page 2.

Comment 2 – BPS as heuristic vs post hoc explanatory tool

The reviewer suggested clarifying whether examples illustrate prospective hypothesis generation or retrospective interpretation.

Response

I appreciate this helpful suggestion. I have added clarifying language explaining that the biopsychosocial model may function both as a prospective multilevel heuristic guiding hypothesis generation and as a retrospective framework organizing empirical findings. The paragraph now explicitly states that the model encourages formulation of multilevel hypotheses tested longitudinally.

Location of change: Section “*Achievements: When the Model Delivers*”, pp. 2-3:

Importantly, in some of these domains the BPS model explicitly motivated integrative hypotheses, whereas in others it has served as a retrospective framework to organize converging findings. Its heuristic value lies not in claiming priority for any level, but in encouraging researchers to formulate multilevel hypotheses a priori, for example by specifying how stress exposure, inflammatory processes, and relational context jointly predict disease course, and by testing such models longitudinally. Used in this way, the BPS model becomes a research strategy rather than a post hoc interpretive lens.

Comment 3 – Inflammation and depression; bidirectionality

The reviewer emphasized that inflammation should not be presented as having causal primacy in depression.

Response

I agree and revised the paragraph accordingly. The text now explicitly states that inflammation should not be granted causal primacy but rather acts as a mediator linking social stress, psychological processes, and disease. I also added references discussing bidirectional stress–inflammation models (Berk et al., Slavich & Irwin, Watt).

Location of change: Section “*Achievements: When the Model Delivers*”, page 2:

Moreover, social stressors and depressive states themselves can trigger inflammatory responses, thus inflammation should not be granted causal primacy but viewed as a key mediator linking social context, psychological processes, and disease.¹³⁻¹⁵ The bidirectional causality, where depression can both result from and drive inflammation, has been central to recent debates in psychoneuroimmunology.¹⁶

Comment 4 – Realism of defining causal pathways

The reviewer questioned how realistic it is to expect researchers to rigorously define causal pathways given the complexity of psychiatric disorders.

Response

I agree and have softened the language. The text now clarifies that specifying causal pathways does not imply identifying a single decisive mechanism but rather formulating provisional multilevel causal schemas that can be empirically tested and revised over time.

Location of change: Section “*Critiques: Missing the Mark*”, page 3:

In practice, particularly for disorders like depression, “rigorously defining” causal pathways does not mean isolating a single decisive mechanism, but formulating provisional, multilevel causal schemas that can be empirically tested, revised, and constrained over time.

Comment 5 – Graph theory and network analysis

The reviewer suggested mentioning network analysis as a promising modeling approach.

Response

I have incorporated network approaches and graph-theoretical modeling as complementary tools for capturing multilevel dynamics over time, with reference to Borsboom & Cramer.

Location of change: Section “*The Next Fifty Years: Causal Integration Ahead*”, page 5:

Network approaches and graph-theoretical modeling further allow estimation of dynamic relationships among symptoms, behaviors, and biological variables over time, capturing how feedback loops stabilize or destabilize health states.⁵²

Comments from Reviewer #2

Comment 1 – Engagement with integrative frameworks (CES, enactivism, etc.)

The reviewer recommended engaging more directly with integrative frameworks such as cultural-ecosocial systems (CES), enactivism, and related work.

Response

I thank the reviewer for this important suggestion. I have added a paragraph situating my proposal alongside recent integrative approaches including active inference, enactivism, externalist psychiatry, and the cultural-ecosocial systems framework. These approaches are now explicitly presented as complementary efforts aimed at causal integration rather than positions being dismissed.

Location of change: Section “*Critiques: Missing the Mark*”, pp. 3-4:

In the same spirit, recent work within active inference,³⁸ enactivism,³⁹ and philosophical psychiatry⁴⁰ has called for a more causally integrated understanding of the BPS model. Approaches such as symptom inference models, externalist psychiatry, and the cultural-ecosocial systems framework foreground subjectivity, narrativity, and cultural processes as constitutive of mechanisms rather than contextual add-ons.⁴¹ These efforts are complementary to the present proposal, whose goal is not to replace such frameworks, but to emphasize shared commitments: explicit modeling of cross-level causation and empirical tractability.

Comment 2 – Mechanism vs meaning distinction

The reviewer noted that the phrase “distinguishing mechanisms from meaning” risks reinstating a problematic dualism.

Response

I agree and removed this formulation. The revised text now describes philosophy’s role more modestly as clarifying conceptual assumptions and explanatory levels without replacing empirical inquiry.

Location of change: Section “*The Next Fifty Years: Causal Integration Ahead*”, page 5:

As psychiatry becomes increasingly measurable, philosophy can illuminate conceptual assumptions, clarify explanatory levels, and refine definitions, without substituting for empirical inquiry. Its value lies in clarifying reasoning without competing with science. Such discipline is essential if psychiatry is to remain aligned with medicine⁴⁸

Comment 3 – Overstated claims about quantification and causal mapping

The reviewer suggested moderating claims regarding the ability of current methods to quantify interactions or map causal architecture.

Response

I have revised the language accordingly. Phrases suggesting that contemporary tools “map causal architecture” or fully “quantify interactions” have been replaced with more cautious wording indicating that these methods enable modeling and partial empirical testing of cross-level interactions.

Location of change: Abstract and sections “Critiques” and “The Next Fifty Years”.

Comment 4 – Characterization of Kandel

The reviewer noted that Kandel should not be portrayed as a reductionist.

Response

I agree and revised the description. Kandel is now described as emphasizing biological embedding of psychosocial processes rather than denying psychosocial causation.

Location of change: Section “*Critiques: Missing the Mark*”, page 3:

Some biological integrationists emphasize that psychosocial interventions exert effects through measurable neural plasticity, highlighting biological embedding rather than denying psychosocial causation.²⁸

Comment 5 – Characterization of phenomenology and enactivism

The reviewer suggested revising descriptions of phenomenological and enactivist approaches.

Response

I revised the text to clarify that these perspectives emphasize embodied and relational processes as constitutive of biological dynamics, rather than simply reversing explanatory hierarchies.

Location of change: Section “*Critiques: Missing the Mark*”, page 3:

Phenomenological and enactivist scholars argue that lived experience and relational context are constitutive of biological processes rather than secondary to them.²⁹

Comments from Reviewer #3

Comment 1 – Underrepresentation of social factors

The reviewer noted that examples in the manuscript emphasized biological and psychological processes more than social determinants.

Response

I agree and strengthened the role of social determinants in several ways:

- added examples of structural social influences such as social integration and neighborhood deprivation affecting biological outcomes
- clarified that social factors may exert biological effects independently of psychological mediation
- expanded discussion of structural determinants in the future implementation section
- extended the clinical vignettes in Table 2 to include social and structural factors and corresponding interventions.

Location of changes: Section “*Achievements: When the Model Delivers*”, page 2:

Social influences also operate at a structural level and can shape biological outcomes directly, independent of individual psychological mediation. In a large population-based cohort, low social integration and limited financial support predicted incident cardiovascular events and mortality even after adjustment for socioeconomic indicators, health behaviors, biological risk factors, and depression.²⁴ In a population-based diabetes cohort, neighborhood deprivation predicted cardiometabolic risk, including higher body mass index and poorer glycemic control, independent of individual-level income and education.²⁵

Section “*The Next Fifty Years: Causal Integration Ahead*”, pp. 5-6:

A causal understanding does not automatically translate into better care; implementation requires translation into decision-making algorithms, training, and resource allocation. However, a causally informed framework can improve practice by distinguishing primary drivers from downstream correlates and mediators within a given case. This enables prioritization of upstream biological, psychological, and social determinants, sequencing of interventions according to modeled interdependencies, and avoidance of redundant or low-yield treatments. Rather than adding biological, psychological, and social measures in parallel, the aim is to intervene where cross-level effects are most likely to propagate.

Comment 2 – Multidisciplinary adoption

The reviewer suggested emphasizing that implementing the BPS framework requires collaboration across disciplines.

Response

I agree and added explicit language stating that causal integration requires collaboration across medicine, psychology, neuroscience, public health, and social sciences.

Location of change: Section *“The Next Fifty Years: Causal Integration Ahead”*, page 5:

Fifty years after Engel’s original call, the task is no longer defending the BPS model against biological or psychosocial dogmatism but to rebuild it scientifically by uniting medicine, psychology, neuroscience, public health, and social sciences through cross-level causal modeling in research and training. . Without such cross-disciplinary adoption, the BPS framework risks remaining aspirational.

Comment 3 (A) – “Translates into biological risk”

The reviewer recommended replacing the ambiguous phrase.

Response

The sentence has been revised to clarify that perceived stress prospectively predicts and is associated with measurable biological risk markers.

Location of change: Section *“Achievements: When the Model Delivers”*, page 2:

A longitudinal study showed that perceived stress predicted amygdalar activation, bone marrow stimulation, arterial and systemic inflammation, and subsequent cardiovascular events,²⁰ illustrating how subjective experience can prospectively predict and be associated with measurable biological risk.

Comment 4 (B) – “Laboratories for its evolution”

The reviewer suggested revising the phrase describing mental disorders as “laboratories.”

Response

The sentence has been rewritten to clarify that functional and mental disorders highlight multilevel interactions rather than serving as metaphorical laboratories.

Location of change: Section *“Critiques: Missing the Mark”*, page 3:

Far from exposing the BPS model’s limits, such discoveries show its refinement: functional and mental disorders are not failures of the BPS framework but highlight precisely the kinds of multilevel interactions it was designed to address

Comment 5 (C) – “Brain as secondary” section

The reviewer raised concerns about the philosophical framing and suggested avoiding polemical arguments.

Response

I substantially revised this section. The discussion now focuses on the brain as one dynamically embedded node within a multilevel causal system, without framing the discussion as a critique of specific philosophical positions.

Location of change: Section *“Lessons: Misapplications of the Model”*, page 4:

This emphasis on causal modeling serves all medicine (not just psychiatry), specifying how social stressors predict surgical infections or distress accelerates cancer, without biological reductionism. In fact, some enactivist and relational approaches emphasize that cognition and experience emerge through dynamic interactions between brain, body, and environment, challenging the view that the brain alone generates mental states.⁴² Rather than opposing such

perspectives, a causal BPS framework can treat the brain as one dynamically embedded node within a larger system. Neuroimaging and interventional studies, including sham-controlled transcranial magnetic stimulation trials, show that modulation of neural circuits can alter mood,⁴³⁻⁴⁵ while environmental and relational changes likewise reshape neural function.^{46,47} The implication is not that one level holds causal priority, but that causal influence is distributed and reciprocal across levels. The clinical task is therefore not to privilege the brain, nor to sideline it, but to understand how neural, psychological, and social processes co-constitute one another over time. The limited translational yield of reductionist biological psychiatry over past decades underscores the need for integration rather than abandonment of biology.

Comment 6 (D) – Libet experiments

The reviewer suggested removing the Libet discussion.

Response

I agree and removed the entire paragraph discussing Libet experiments and the reducibility of intention.

Location of change: Section “*Lessons: Misapplications of the Model*” (paragraph removed).

Comment 7 (E) – Mechanism vs meaning

The reviewer proposed to omit the part “distinguishing mechanisms from meaning”, as it is not crucial for understanding the sentence.

Response

As noted above in my response to comment 2 from reviewer #2, I removed this formulation. The revised text now describes philosophy’s role more modestly as clarifying conceptual assumptions and explanatory levels without replacing empirical inquiry.

Location of change: Section “*The Next Fifty Years: Causal Integration Ahead*”, page 5:

As psychiatry becomes increasingly measurable, philosophy can illuminate conceptual assumptions, clarify explanatory levels, and refine definitions, without substituting for empirical inquiry. Its value lies in clarifying reasoning without competing with science. Such discipline is essential if psychiatry is to remain aligned with medicine⁴⁸

Comments from Reviewer #4

Comment 1 – Positioning within broader causal reformulations

The reviewer recommended situating the manuscript within existing causal reformulations of the BPS model.

Response

I agree and expanded the discussion to situate the manuscript alongside active inference, enactivism, externalist psychiatry, and cultural-ecosocial systems approaches, acknowledging shared goals of causal integration.

Location of change: Section “*Critiques: Missing the Mark*”, pp. 3-4:

In the same spirit, recent work within active inference,³⁸ enactivism,³⁹ and philosophical psychiatry⁴⁰ has called for a more causally integrated understanding of the BPS model. Approaches such as symptom inference models, externalist psychiatry, and the cultural-ecosocial systems framework foreground subjectivity, narrativity, and cultural processes as

constitutive of mechanisms rather than contextual add-ons.⁴¹ These efforts are complementary to the present proposal, whose goal is not to replace such frameworks, but to emphasize shared commitments: explicit modeling of cross-level causation and empirical tractability.

Comment 2 – Gap between causal understanding and clinical benefit

The reviewer noted that the manuscript did not sufficiently explain how causal modeling improves clinical practice.

Response

I added a paragraph explicitly discussing the **translation of causal modeling into clinical decision-making**. The text now explains how causal frameworks might:

1. Identify **leverage points for intervention**,
2. Reduce **redundant or low-yield treatments**, and
3. Enable **stratification of interventions across biological, psychological, and social levels**.

Location of change: Section “*The Next Fifty Years: Causal Integration Ahead*”, pp. 5-6:

A causal understanding does not automatically translate into better care; implementation requires translation into decision-making algorithms, training, and resource allocation. However, a causally informed framework can improve practice by distinguishing primary drivers from downstream correlates and mediators within a given case. This enables prioritization of upstream biological, psychological, and social determinants, sequencing of interventions according to modeled interdependencies, and avoidance of redundant or low-yield treatments. Rather than adding biological, psychological, and social measures in parallel, the aim is to intervene where cross-level effects are most likely to propagate.

Comment 3 – Structural determinants and politics

The reviewer emphasized the importance of addressing structural determinants.

Response

We agree and added explicit discussion of structural determinants including **poverty, discrimination, climate stress, migration, transportation, and sanitation**, emphasizing that addressing these requires policy-level interventions.

Location of change

Section “**The Next Fifty Years: Causal Integration Ahead**”, paragraph discussing structural determinants.

Comment 4 – Characterization of relational and enactivist approaches

The reviewer notes that relational and enactivist approaches may be understood as interpretive frameworks for mechanisms rather than as alternatives to mechanistic explanation.

Response

I agree and revised the passage to avoid suggesting that these perspectives lack mechanistic relevance. The text now presents enactivist and relational approaches as compatible with causal biopsychosocial modeling of distributed brain–body–environment interactions. This revision aligns with changes made in response to Reviewer 2 (comment on engagement with integrative frameworks such as CES, enactivism, and active inference) and Reviewer 3 (comment 5 regarding the “brain as secondary” section).

Comment 5 – Role of philosophy and “distinguishing mechanisms from meaning”

The reviewer asked for clarification of the role of philosophy and the meaning of the phrase “distinguishing mechanisms from meaning.”

Response

I agree and simplified this passage. The sentence has been revised to clarify that philosophy contributes by clarifying conceptual assumptions and explanatory levels, while empirical research determines mechanisms. The phrase “distinguishing mechanisms from meaning” has been removed. This revision also addresses concerns raised by Reviewer 2 (comment on mechanism–meaning distinction) and Reviewer 3 (comment 7).